# Size, demography, ownership profiles, and identification rate of the owned dog population in central Italy

**Andrea Carvelli**[1]*, **Paola Scaramozzino**[1], **Francesca Iacoponi**[1¤], **Roberto Condoleo**[1], **Ugo Della Marta**[2]

**1** Epidemiology Unit, Istituto Zooprofilattico Sperimentale del Lazio e della Toscana "M. Aleandri", Via Appia Nuova, Roma, Italy, **2** Istituto Zooprofilattico Sperimentale del Lazio e della Toscana "M. Aleandri", Via Appia Nuova, Roma, Italy

¤ Current address: Department of Food Safety, Nutrition and Veterinary Public Health, Istituto Superiore di Sanità, Viale Regina Elena, Roma, Italy.
* andrea.carvelli@izslt.it

**Data Availability Statement:** All relevant data are within the manuscript and its Supporting Information files.

## Abstract

The One Health paradigm recognizes that information on infectious diseases, zoonosis and related predictors in animal populations is essential. Pets live in close contact with humans and interact with wild animals, but the lack of reliable information on pet population size, demography and ownership profiles is a constant worldwide. Reliable data must be made available in order to address proper public health policies regarding the design of surveillance plans, the management of canine welfare and stray dog phenomenon, the control of dog behaviour-related problems, and the livestock/wildlife endangerment. Dog identification & registration (I&R) have become mandatory in most European countries in recent years, but the process is far from being widely accomplished, thus resulting in an underestimation of the real canine population. To date, data on the completeness of Dog Registries is very limited. A cross-sectional survey through 630 face-to-face questionnaires was performed with the aim of investigating the dog population size, demography, ownership profiles, and the I&R rate in central Italy. Logistic regression models investigated risk factors with the following outcome variables: dog presence into the Dog Registry, veterinary care frequency, and dog ownership. The present study identified that the dog population is higher than previously reported in Italy and in Europe, whilst lower compared to countries with a poor Human Development Index (a statistic composite index of life expectancy, education, and per capita income indicators used by United Nation). Almost half of the interviewed people (47%) owned at least one dog, totalling 315 dogs. The mean number of dogs per household was 0.7 and 1.5 per dog-owning household. The Bayesian model estimated 481,294 (95% CI: 470,860–491,978) dogs (90 dogs/km$^2$; human:dog ratio 9.0, 95% CI: 8.8–9.2) suggesting that a high density of people does not limit the dog ownership. Dogs resulted predominant among pets. The majority of dogs were males, crossbred, acquires as a gift, lived in rural areas, outdoors and attended a veterinary visit 1–2 times per year. The percentage of neutered dogs were higher in females (55%) than in males (8%). Only 75.3% (95% CI: 73.6–76.9) of the dogs were correctly identified and registered. The Dog Registry completeness

**Funding:** The author(s) received no specific funding for this work.

**Competing interests:** The authors have declared that no competing interests exist.

increased during the first decades after its establishment, but no improvement has been made afterwards. The dogs correctly identified and registered were more likely to be pure-bred, neutered, lived in urban areas and visited a veterinarian frequently. Several strategies are recommended to encourage I&R, including promoting responsible dog ownership, engaging private veterinarians and dog breeders, and establishing an effective control system. The present study identified also that the dog source and the kind of feeding were variables associated with the veterinary care frequency. Owning a dog was associated with living in rural areas and the presence of children. The present study reported the poor presence of reliable predictors for the dog ownership. This is the first study to provide an estimate of the canine population abundance, characteristics, and ownership profiles in a European large metropolitan area.

## Introduction

The One Health paradigm recognizes that standardized and complete information on infectious diseases, zoonoses and risk factors related to their spatial and temporal distributions in pet population is essential [1]. Pet dogs and cats live in close contact with humans and can interact with other pets and wild animals. Therefore, the knowledge of animal populations in terms of size, demographic characteristics, and ownership profiles is a key tool of the whole process. While in livestock species rules on the identification, the registration and related information on abundance, descriptive records (sex, age, breed, owners, etc) and informative systems to manage these data have been developed on a large-scale basis, the lack of information in pet populations is a constant worldwide.

### Canine population size

Several papers have been published on owned dog population estimates. Some studies measured the human:dog ratio (HDR), some estimated the percentage of dog-owning households (HH) or the mean number of dogs per HH and others calculated the regression coefficient for predictors [2]. The World Health Organization (WHO) estimated a HDR between 6 and 10 in American and European countries and a ratio varying from 7 to 10 in 'developing countries' [3, 4]. Pets are widely present in European HH and their numbers show a growing trend. The number of dogs estimated by the pet food industry in Europe is 85,184,000 [5]. The same source assessed 7,002,000 pet dogs in Italy (i.e. HDR 8.6), while the Italian Ministry of Health recorded 11,753,180 dogs (i.e. HDR 5) [6].

### Identification & registration rate: Dog Registry

In most European countries, dog identification and registration (I&R) have become mandatory in recent years [7]. Nevertheless, while identification is performed on an individual basis, registration in a central accessible database (a Dog Registry–DR) is not compulsory in some countries or not implemented effectively in others. Consequently, the completeness of DRs is far from being achieved [7] and little data are published on this topic. To date, reliable data on the dog population abundance are provided by industry or *ad hoc* surveys rather than managed within an institutional framework [1]. In Italy, since 1991, every owned dog and stray dog caught by the competent authorities must be identified by an electronic transponder (microchip) and registered in a DR (Framework Law 281/1991). The main aim of the law was

to contrast the phenomenon of abandoning dogs. Both private and public veterinarians have access to the DR and they can visualize or manage data of both the animals and owners (e.g. sex, age, breed, address, owner, transfer, date of birth/death, etc.). Since the DR establishment, many information campaigns have been organised toward the general population through radio, newspaper, social media, leaflet and advertising posters. The I&R costs are affordable (about 15€).

The accuracy of the DR is uncertain because there is no active control on dog I&R. Most of stakeholders think that the only purpose of the DR is the reunification of lost dogs with their owners, underestimating the epidemiological relevance of population abundance assessment. The rabies vaccination campaigns in 2008 in Veneto Region (north Italy) and the public health activities in the 2016 earthquake emergency (central Italy) indicated that many dogs were not registered and not identified [8]. A previous study carried out to assess the completeness of the DR in a densely populated area of Rome found that only 75% of the dogs were registered [9]. These findings show that the real owned population is likely underestimated in DRs. Therefore, a periodic assessment of the DR completeness is appropriate and factors influencing owners to register a dog are worth investigating.

## Canine population demography

Information on demography of dog population are underprovided despite the relevance in public health and industrial fields. Demography is essential for the investigation, management and control of several disease and health issues to be addressed by private veterinarians and public health authorities. The dog breed and the living environment can represent risk factors for the susceptibility of transmissible diseases and degenerative or neoplastic diseases [10]. These factors can be supposed to be related to aggressive behaviour responsible of the dog biting phenomenon and its consequences, such as human and animal injuries and rabies vaccination [11]. Furthermore, the role of dogs as sentinel animals for environmental hazards is particularly important in comparative medicine [12]. The age structure of the dog population can be useful for the private sector to predict which health and commercial services to provide and for public health authorities to predict costs of surveillance and control plans. Additionally, there is very little published information about pet veterinary care. Determinants of the veterinary visit frequency need to be assessed and evaluated, because pet access to veterinary assistance is an essential factor in improving animal health and welfare.

## Ownership profiles

A few studies reported reliable predictors for the dog ownership worldwide and there is only one published study in Italy [13–17]. The responsible ownership, defined by the World Animal Health Organisation (OIE) as "*the situation whereby a person accepts and commits to perform various duties according to the legislation in place and focused on the satisfaction of the behavioural, environmental and physical needs of a dog and to the prevention of risks (aggression, disease transmission or injuries) that the dog may pose to the community, other animals or the environment*" has been recognised by the WHO and the OIE as the main strategy to manage pet population health and welfare [3, 18]. Therefore, collecting data on the owners' attitude and characteristics has an impact on strategies that can be applied to monitor and enhance individual and community animal health.

The aims of this study were to estimate the canine owned population size, to evaluate the rate of identification and registration into the DR, to study the demography of the owned dog population, and to identify the human factors influencing the dog ownership.

## Materials and methods

### Study area and survey

The study area was Rome province, covering 5,363 $Km^2$ with a population of 4,331,856 inhabitants (population density: 807.7/$Km^2$). It consists of a metropolis (Rome, 2,839,042 inhabitants) and 120 municipalities surrounded by farmed lands and wooded areas located between the coast and inland territories [19]. The average elevation is 269 meters above sea level and the annual average temperature is 14.5° C. The use of the land was mainly (49%) aimed at agricultural activities, 37% was covered by forest and pastures, 10% was represented by urban area and 4% was classified as another use [20].

Questionnaires data were collected in a District of the Rome province called "Rome 6". It covers 726.7 $Km^2$ with a population of 539,445 inhabitants. It is located between the Tyrrhenian coast and inland territories. The survey was carried out during weekdays between the months of July and December 2013 in the waiting rooms of 4 General Health Care Centres of the National Health System. A systematic sampling was performed by selecting one in every four person amongst the general diagnostic patients. Survey participation was requested within the framework of a research project on human and animal tumours managed by the Ministry of Health. More details were described in a published study [21].

Two different sample sizes were calculated for the different aims of the research. To calculate the required sample size for the evaluation of the DR, an expected prevalence of 75%, a standard error of 5% and a confidence level of 90% were assumed, resulting in 203 questionnaires to be administered to people owning at least one dog. The expected prevalence was intended as the percentage of dog correctly identified and registered into the DR, which resulted 75% in a study performed in a close area [9].

The required sample size for the study of the dog demography and ownership profiles was 664, assuming a 5% of standard error, an unknown prevalence (i.e. 50%) and a 99% of confidence level.

### Questionnaire design

A cross-sectional survey on pet ownership through a face-to-face questionnaire was performed. A detailed description of the survey methodology is provided in a previously published study and in the supporting information [21]. Each person selected for the interview was asked whether they resided in the study area and then provided with information on the survey. The participants were asked whether they owned a dog and/or other pets, the number of owned pets, the characteristics of the owned dog (sex, neutering, breed, age), the source of the dog (born in house, found, gift, adopted from a shelter, purchased), the kind of environment the dog usually lived in (urban/rural area, mainly outdoor/indoor), the kind of feeding (homemade, commercial, mixed), the annual average veterinary care frequency (never, 1–2 times a year, 3 or more times), the number of family members, the presence of children, and the status of identification and registration of the dog into the DR. The final section of the questionnaire collected the personal information of the participant (gender, age, marital status, education level, occupation, habitat–i.e. urban/rural area). In order to have a conservative approach, the presence into the DR was considered as a dichotomous variable where the answer "do not know" was considered as "no". The definition of the dog ownership was based on the respondent's definition. Interviewers emphasized the care provided in terms of regular feeding and health status. The physical restriction of the animal was not taken into consideration.

## Data analysis

The information gathered during the survey was entered into an *ad hoc* Microsoft Access® database (Microsoft Office® 2003). All variables were reported as absolute frequency and percentage (%).

A Bayesian approach was adopted to estimate the size of the canine population based on dogs registered in the DR. Based on data from a similar previous investigation [9], a beta distribution ($\alpha_1 = 287$, $\alpha_2 = 98$) was used as *a prior* to determine the probability that a dog was correctly registered, while the likelihood function was modelled using the data from the present study as a binomial distribution (s = 183, n = 315). Since the beta prior is conjugate to a binomial likelihood, the resulting posterior distribution was a beta distribution with parameters $\alpha'_1 = \alpha1 + s$ and $\alpha'_2 = \alpha2 + n-s$, respectively [22]. Then, we used a simple model to calculate the mean and 95% confidence intervals (CI) of the dog population size in Rome province:

$$N_{dogs} = N_{reg} + N_{reg}(1 - P_{reg})$$

where $P_{reg}$ is a random value generated from a beta distribution with the mentioned parameters (R 3.6.1, rBeta 2009 v 1.0 package, 100,000 iterations) and $N_{reg}$ is the number of registered dogs in the DR (362,277).

Data regarding the DR were collected from the Regional DR Database [23]. In order to eliminate the bias of possible dogs deaths not reported, the animals over the age of 16 years were not counted.

The effect of the exposure variables on three outcome variables (presence of a dog into the DR, frequency of veterinary visit and owning of a dog) was analysed using logistic regression models. Risk factors with a bivariate p value ≤0.25 were included in a multivariable stepwise logistic model [24]. Chi square test and phi coefficient were used to evaluate a possible multicollinearity among risk factors that were significantly associated with the outcomes. The absence of plausible interactions among variables were tested in all regression models. In logistic regression analyses with DR and veterinary care considered as outcome variables (Tables 2 and 3), the HH was included in order to ensure the independence of observations (i.e. an owner owning more than one dog having the same behaviour or attitude with all the dogs). In the logistic regression analysis regarding dog ownership profiles (Table 4), the outcome variable was represented by the owner and not by the dog, avoiding the overrepresentation of HHs with more than one dog.

The likelihood ratio chi square test (LRT) for goodness of fit was calculated to compare the final model with full models. A two-tailed p value <0.05 was considered statistically significant. All statistical analyses were performed by Stata/SE version 12 for Windows (StataCorp LP, TX, USA).

## Results

A total of 630 people were interviewed, the answer rate was 76% and 455 people were considered in the analyses because they resided in the study area. The total number of interviewed people and their family members was 1,297. The mean number of persons per HH was 2.9, higher than the Italian mean, 2.4 [19].

Overall, 244 (53%) interviewed people did not have dogs while 212 (47%) owned at least one dog. Of these, 140 (66%) owned one dog, 53 (25%) two dogs, 13 (6%) three dogs, 6 (3%) four or more dogs, totalling 315 dogs. Among these, 183 (58%) were correctly identified and registered into the DR. The mean number of dogs per HH was 0.7 and 1.5 per dog-owning HH.

At the time of the survey, 362,277 dogs were registered in the DR ($N_{reg}$). The model estimated that the mean number of dogs relative to the province of Rome was 481,294 (95% CI: 470,860–491,978), 90 dogs/km$^2$ and the HDR was 9.0 (95% CI: 8.8–9.2). Considering this estimate, 75.3% (362,277/481,294; 95% CI: 73.6–76.9) of the dogs living in Rome province were correctly registered into the DR.

The overall median age was 5.0 years (Q1 = 2.0; Q3 = 8.0) and it was similar in males (median = 5.0; Q1 = 2.0; Q3 = 8.8) and females (median = 5.0; Q1 = 2.0; Q3 = 8.0). The age distribution of the dog population showed a moderately different shape for sex (Fig 1).

The majority of dogs were males (52%; male:female ratio: 1.1), not neutered, crossbred, acquired as a gift, living in a rural area, mainly outdoors, fed mixed food, attended a veterinary visit 1–2 times per year and were registered in the DR (Table 1). The percentage of neutered animals was 8% and 55% of males and females, respectively.

The descriptive analysis showed that the majority of dogs registered in the DR were female, adult, purebred, acquired as a gift, not neutered, fed commercial food, lived in a rural area mainly outdoors, attended a veterinary visit at least once a year, lived in a family with 3 or more members and without children.

The registration into the DR was considered to explore its potential association with exposure variables (Table 2). A significant association was found for sex and kind of feeding in the univariable analysis. In the multivariable model, findings revealed that being a purebred dog, neutered, with intensive veterinary care, and living in an urban area were predictors of the presence into the DR. HH was found to have no effect on the outcome variable (p = 0.07).

Taking into consideration the average veterinary care, the descriptive analysis showed that dogs visiting a veterinarian at least once a year were mainly males, adult, crossbred, acquired as a gift, not neutered, lived in rural areas and outdoors, registered in the DR and commonly fed commercial or mixed food (Table 3). In the logistic regression analyses, a significant association was found for the breed and the presence into the DR in the univariable analysis while multivariable model confirmed the dog source and the kind of feeding as predictors. HH was found to have no effect on the outcome variable (p = 0.432).

Exploring the ownership profiles, the descriptive analysis showed that the majority of dog owners were females, married or long term committed, with a middle educational level,

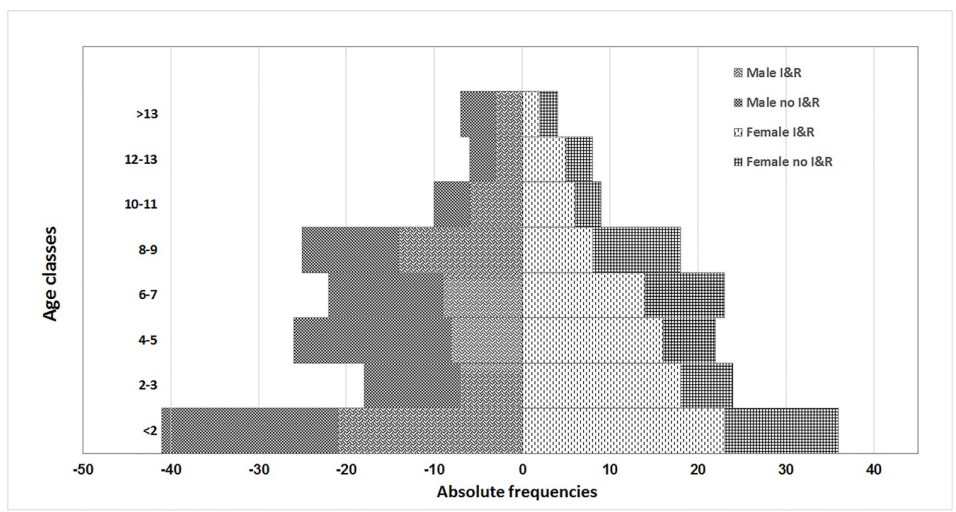

**Fig 1. Dog population pyramid by sex, identification and registration (I&R) into the Dog Registry.**

**Table 1. Demographic characteristics of enrolled dogs based on a survey in central Italy in 2013.**

| | | N | % | Missing |
|---|---|---|---|---|
| Sex | Male | 164 | 52.1 | 0 |
| | Female | 151 | 47.9 | |
| Overall neutering | No | 219 | 69.5 | 0 |
| | Yes | 96 | 30.5 | |
| Neutered females | No | 68 | 45.0 | |
| | Yes | 83 | 55.0 | |
| Neutered males | No | 151 | 92.1 | |
| | Yes | 13 | 7.9 | |
| Breed | Crossbred | 164 | 53.2 | 8 |
| | Purebred | 144 | 46.8 | |
| Source | Born in house | 17 | 5.5 | 6 |
| | Found | 64 | 20.7 | |
| | Gift | 148 | 47.9 | |
| | Adopted | 28 | 20.7 | |
| | Purchased | 52 | 16.8 | |
| Habitat | Urban area | 113 | 56.8 | 4 |
| | Rural area | 199 | 63.8 | |
| Living environment | Indoors | 75 | 35.2 | 28 |
| | Outdoors | 213 | 74.0 | |
| Feeding | Homemade | 22 | 7.1 | 3 |
| | Commercial | 117 | 37.5 | |
| | Mixed | 173 | 55.4 | |
| Veterinary visit | Never | 29 | 9.3 | 3 |
| | 1–2 times | 178 | 57.1 | |
| | 3 or more times | 105 | 33.7 | |
| Dog Registry | Microchip | 172 | 55.1 | 3 |
| | Tattoo | 10 | 3.2 | |
| | No | 100 | 32.1 | |
| | Do not now | 30 | 9.6 | |

working from home, lived in a numerous family, in a rural area and had no other pets. Of the 212 people owning a dog, 31% also owned a cat and 9% owned a pet different from a cat, such as birds, fish or turtles.

The ownership of a dog was then considered to explore its potential association with exposure variables (Table 4). A significant association was found with the presence of a cat in the univariable analysis. In the multivariable model, habitat and the presence of children were confirmed as predictors of the dog ownership.

There was no multicollinearity between the independent variables. In the regression models, interaction terms were not significant.

## Discussion

The present study investigated the dog population size, demography, ownership profiles and the I&R rate in the biggest Italian metropolitan area. Reliable data must be available and regularly updated to address proper policies by competent authorities in a public health concept. The prevention of zoonotic disease including rabies, leishmaniasis, echinococcosis, the design of surveillance plans, the management of livestock predation, wildlife endangerment, owned-

**Table 2. Descriptive, univariable and multivariable logistic analysis of characteristics associated with dogs registered in the Dog Registry in central Italy in 2013.**

| | Dog Registry | | | |
| --- | --- | --- | --- | --- |
| | No/Do not know (N = 182)(%) | Yes (N = 130)(%) | Univariable analysis OR (95% CI) | Multivariable model OR (95% CI) |
| **Sex** | | | | |
| Male | 78 (60) | 86 (47) | - | |
| Female | 54 (40) | 96 (53) | 1.6 (1.0–2.6)* | |
| *Missing* | 1 | 0 | | |
| **Age (years)** | | | | |
| ≤2 | 36 (29) | 43 (24) | - | |
| 2.1–8 | 52 (42) | 99 (56) | 1.5 (0.9–2.7) | |
| >8 | 37 (30) | 34 (19) | 0.7 (0.4–1.4) | |
| *Missing* | 6 | 6 | | |
| **Breed** | | | | |
| Crossbred | 78 (61) | 83 (47) | - | - |
| Purebred | 50 (39) | 93 (53) | 1.7 (1.1–2.8)* | 1.7 (1.0–2.8)* |
| *Missing* | 2 | 6 | | |
| **Source** | | | | |
| Born in house | 7 (6) | 10 (6) | - | |
| Found | 31 (25) | 33 (18) | 0.7 (0.3–2.2) | |
| Adopted | 9 (7) | 18 (10) | 1.5 (0.4–5.2) | |
| Gift | 68 (54) | 77 (43) | 0.8 (0.3–2.2) | |
| Purchased | 10 (8) | 42 (23) | 2.9 (0.9–9.6) | |
| *Missing* | 5 | 1 | | |
| **Neutering** | | | | |
| No | 102 (78) | 115 (63) | - | - |
| Yes | 28 (22) | 67 (37) | 2.1 (1.3–3.6)* | 1.9 (1.0–3.3)* |
| *Missing* | 0 | 0 | | |
| **Feeding** | | | | |
| Homemade | 14 (11) | 8 (4) | - | |
| Commercial | 38 (29) | 78 (43) | 3.6 (1.4–9.3)** | |
| Mixed | 77 (60) | 96 (53) | 2.2 (0.9–5.5) | |
| *Missing* | 1 | 0 | | |
| **Veterinary visit** | | | | |
| Never | 18 (14) | 11 (6) | - | - |
| 1–2 times | 89 (69) | 88 (48) | 1.6 (0.7–3.6) | 1.4 (0.6–3.1) |
| 3 or more times | 22 (17) | 83 (46) | 6.2 (2.5–15.0)*** | 4.9 (1.9–12.3)*** |
| *Missing* | 1 | 0 | | |
| **Habitat** | | | | |
| Urban | 33 (26) | 79 (44) | - | - |
| Rural | 96 (74) | 102 (56) | 0.4 (0.3–0.7)*** | 0.4 (0.2–0.7)*** |
| *Missing* | 1 | 1 | | |
| **Living environment** | | | | |
| Indoors | 26 (21) | 49 (30) | - | |
| Outdoors | 95 (79) | 117 (70) | 0.7 (0.4–1.1) | |
| *Missing* | 9 | 16 | | |
| **Family member** | | | | |
| 1 | 4 (3) | 8 (4) | - | |
| 2 | 24 (19) | 33 (18) | 0.7 (0.2–2.5) | |
| ≥3 | 99 (78) | 140 (77) | 0.7 (0.2–2.4) | |

(*Continued*)

**Table 2.** (Continued)

| | No/Do not know (N = 182)(%) | Yes (N = 130)(%) | Univariable analysis OR (95% CI) | Multivariable model OR (95% CI) |
|---|---|---|---|---|
| | | | **Dog Registry** | |
| *Missing* | 3 | 1 | | |
| Children | | | | |
| No | 81 (64) | 114 (63) | - | |
| Yes | 46 (36) | 68 (37) | 1.0 (0.7–1.7) | |
| *Missing* | 3 | 0 | | |

*p<0.05;

**p<0.01;

***p<0.001

LR = -181.68 (p<0.001)

wildlife canine population bond, behaviour-related problems (aggressiveness, barking noise, dog-to-human bite), and the impact on canine welfare, are all issues that can be efficiently driven only if the population size, demography and their trends are known and trustworthy. Furthermore, the size and the demography of the canine population is important for many stakeholders: private veterinarians, pet products manufacturers, insurance companies and researchers [25]. Consistent information is crucial to calculate canine diseases incidence, for the possible use of dogs as sentinel animals (e.g. environmental contaminants) and to predict the future population dynamic.

The lack of reliable, centralised and official information leads to the use of surveys to estimate the dog population. Several methods can be implemented to collect data in cross-sectional procedures: mailed, door-to-door, telephone surveys. Most of them are subject to a representativeness bias. The approach used in the present paper, a face-to-face questionnaire, though costly and time-consuming, improves the precision of the estimates limiting the selection, non-response and the measurement biases [2]. Biases were also minimised recruiting respondents within the framework of a research project on human and animal tumours and not directly on the dog ownership. Interviews were performed to general people in Health Care Centres while attending for the booking, the payment or routine diagnostic exams. As the Italian Health Care System provides comprehensive coverage to all citizens, routine diagnostic exams are performed in the Health Care Centres homogenously among age, sex and socio-economic levels. For these reasons, the sampled population was assumed representative of the general population [21].

This research has some limitations. Rome is a province with a large population and a human density, limiting the generalizability of our findings to the rest of the country. Additionally, the *a priori* probability distribution used in the Bayesian approach was conducted in a limited part of the Rome province [9]. The interviewed people could not be the responsible of the dog management in the HH and this could limit the relation between their characteristics and the dog I&R. Finally, the stray dog population was not considered in this study.

## Canine population size

Different indices are used in pet population studies and they are not consistent with each other. The human to dog ratio, the percentage of HHs owning a dog, the mean dog per HH, the number of dogs per km$^2$ are frequently used, together with capture-recapture or Bayesian

**Table 3. Descriptive, univariable and multivariable logistic analysis of characteristics associated with the dog veterinary care in central Italy in 2013.**

| | Veterinary visit per year | | | |
|---|---|---|---|---|
| | Never (N = 29)(%) | 1 or more times (N = 283) (%) | Univariable analysis OR (95% CI) | Multivariable model OR (95% CI) |
| **Sex** | | | | |
| Male | 13 (68) | 143 (51) | - | |
| Female | 6 (32) | 139 (49) | 1.8 (0.8–4.1) | |
| *Missing* | 0 | 1 | | |
| **Age (years)** | | | | |
| ≤2 | 3 (16) | 73 (27) | - | |
| 2.1–8 | 6 (32) | 141 (52) | 0.9 (0.3–2.6) | |
| >8 | 10 (53) | 59 (22) | 0.4 (0.1–1.1) | |
| *Missing* | 0 | 10 | | |
| **Breed** | | | | |
| Crossbred | 13 (68) | 141 (51) | - | |
| Purebred | 6 (32) | 134 (49) | 2.5 (1.0–5.8)* | |
| *Missing* | 0 | 8 | | |
| **Source** | | | | |
| Born in house | 6 (20) | 11 (4) | - | - |
| Found | 7 (23) | 58 (21) | 5.2 (1.4–19.4)** | 5.8 (1.5–23.0)** |
| Adopted | 5 (17) | 23 (8) | 2.5 (0.6–10.0) | 2.5 (0.6–10.8) |
| Gift | 11 (37) | 135 (49) | 6.7 (2.1–21.6)*** | 7.3 (2.2–24.8)** |
| Purchased | 1 (3) | 51 (3) | 27.8 (3.0–254.9)** | 28.3 (2.9–279.7)** |
| *Missing* | 0 | 5 | | |
| **Neutering** | | | | |
| No | 15 (79) | 194 (69) | - | |
| Yes | 4 (21) | 89 (31) | 1.4 (0.6–3.5) | |
| *Missing* | 0 | 0 | | |
| **Feeding** | | | | |
| Homemade | 4 (24) | 15 (5) | - | - |
| Commercial | 3 (16) | 113 (40) | 13.2 (3.4–50.4)*** | 14.2 (3.5–57.7)*** |
| Mixed | 12 (63) | 155 (55) | 4.0 (1.4–11.2)* | 5.1 (1.7–15.3)* |
| *Missing* | 0 | 0 | | |
| **Habitat** | | | | |
| Urban | 3 (16) | 104 (37) | - | |
| Rural | 16 (84) | 178 (63) | 0.7 (0.3–1.5) | |
| *Missing* | 0 | 1 | | |
| **Living environment** | | | | |
| Indoors | 6 (24) | 69 (26) | - | |
| Outdoors | 19 (76) | 194 (74) | 0.9 (0.3–2.3) | |
| *Missing* | 7 | 20 | | |
| **Dog registry** | | | | |
| No/Do not know | 10 (53) | 111 (39) | - | |
| Microchip/Tattoo | 9 (47) | 171 (61) | 2.5 (1.1–5.5)* | |
| *Missing* | 0 | 1 | | |
| **Family member** | | | | |
| 1 | 0 (0) | 11 (4) | - | |
| 2 | 3 (16) | 54 (19) | 1.6 (0.2–17.2) | |
| ≥3 | 16 (84) | 215 (77) | 0.8 (0.1–6.3) | |
| *Missing* | 0 | 3 | | |

(*Continued*)

**Table 3.** (Continued)

| | Veterinary visit per year | | | |
|---|---|---|---|---|
| | **Never (N = 29)(%)** | **1 or more times (N = 283) (%)** | **Univariable analysis OR (95% CI)** | **Multivariable model OR (95% CI)** |
| Children | | | | |
| No | 9 (47) | 180 (64) | - | |
| Yes | 10 (53) | 101 (36) | 0.6 (0.3–1.3) | |
| *Missing* | 0 | 2 | | |
| Cat | | | | |
| No | 17 (59) | 186 (66) | | |
| Yes | 12 (41) | 97 (34) | 0.7 (0.3–1.6) | |
| *Missing* | 0 | 0 | | |
| Other pet | | | | |
| No | 27 (93) | 251 (89) | | |
| Yes | 2 (7) | 32 (11) | 1.7 (0.4–7.6) | |
| *Missing* | | | | |

*p<0.05;

**p<0.01;

***p<0.001

LR = -80.85 (p<0.001)

methods to infer the absolute population size [26]. In the present studies, all the possible indices were calculated and presented to enhance the precision of the estimates [27] and to help researchers to compare results with other studies worldwide.

Dogs are popular in Italy and the findings of the present study confirmed an extensive presence of the dog among human population. The percentage of respondents having a dog was 47%, much higher than previous findings in Italy (33%, 25%) and in Europe (39%, 31%), but lower if compared to countries with a poor Human Development Index (a statistic composite index of life expectancy, education, and per capita income indicators used by United Nation) (65%, 51%) [14,16,17,28–30]. Interestingly, taking in consideration other indices depicts a different pattern. The findings of the present study denoted a massive presence of dogs with 90 dogs/km$^2$ (34 and 46 in previous studies in central and northern Italy, respectively [17, 28]). In contrast, due to a high human density, the HDR (9.0) was the highest (i. e. a relative low number of dogs) if compared with what is reported worldwide (in some cases calculated by the authors from data present in published studies). A HDR ranging from 3 to 5 was found in Brazil, Mexico and in an Italian Province close to Rome [17, 27, 29]. A HDR from 5.4 to 5.8 was reported in 3 studies performed in northern Italy [28, 31, 32] while a score close to 6.0 in UK and in Guatemala [16, 30]. These findings reported a large number of dogs in an area with a high density of people, suggesting that the dog ownership is very popular and anthropogenic pressure seems not to limit it. The wide base of the pyramid (Fig 1) indicates that the population size will likely not decrease. Surprisingly, the mean dog of dog-owning HH reported in the present study (1.5) was almost identical to that reported worldwide (1.3÷1.6), suggesting the number of dogs in an HH can be a constant [14, 16, 17, 30, 32]. Interestingly, the people owning a dog (47%) were more common than the people owning a cat (29%) in the study area [21]. Our findings confirm the previous reports of dog predominance in companion animals [14, 17, 28, 32].

**Table 4. Descriptive, univariable and multivariable logistic analysis of characteristics associated with dog ownership in central Italy in 2013.**

| | Dog owners | | | |
|---|---|---|---|---|
| | No (N = 243)(%) | Yes (N = 212)(%) | Univariable analysis OR (95% CI) | Multivariable model OR (95% CI) |
| Gender | | | | |
| Male | 104 (45) | 78 (37) | - | |
| Female | 128 (55) | 133 (63) | 1.4 (0.9–2.0) | |
| *Missing* | 11 | 1 | | |
| Age (years) | | | | |
| ≤19 | 3 (1) | 2 (1) | - | |
| 20–29 | 18 (8) | 24 (11) | 2.0 (0.3–13.2) | |
| 30–39 | 35 (15) | 34 (16) | 1.4 (0.2–9.3) | |
| 40–49 | 48 (20) | 52 (25) | 1.6 (0.3–10.1) | |
| 50–59 | 55 (23) | 58 (27) | 1.5 (0.2–9.8) | |
| ≥60 | 80 (33) | 42 (20) | 0.8 (0.1–4.9) | |
| *Missing* | 4 | 0 | | |
| Marital status | | | | |
| Single/Separated/Widowed | 71 (31) | 68 (33) | - | |
| Married | 157 (69) | 138 (67) | 0.9 (0.6–1.4) | |
| *Missing* | 15 | 6 | | |
| Education level | | | | |
| Primary school | 21 (9) | 20 (10) | - | |
| Middle school | 68 (31) | 60 (30) | 0.9 (0.4–1.8) | |
| High school | 110 (50) | 89 (45) | 0.8 (0.4–1.7) | |
| University | 22 (10) | 29 (15) | 1.4 (0.6–3.1) | |
| *Missing* | 22 | 14 | | |
| Occupation | | | | |
| Home working | 122 (53) | 89 (43) | - | |
| Office | 74 (32) | 77 (38) | 1.4 (0.9–2.2) | |
| Other | 34 (15) | 39 (19) | 1.5 (0.9–2.7) | |
| *Missing* | 13 | 7 | | |
| **Habitat** | | | | |
| Urban area | 129 (65) | 93 (45) | - | - |
| Rural area | 70 (35) | 116 (55) | 2.3 (1.5–3.4)[***] | 2.1 (1.5–3.3)[***] |
| *Missing* | 44 | 3 | | |
| Family member | | | | |
| 1 | 16 (8) | 10 (5) | - | |
| 2 | 73 (36) | 44 (21) | 1.0 (0.4–2.3) | |
| ≥3 | 112 (56) | 154 (74) | 2.2 (1.0–5.0) | |
| *Missing* | 42 | 4 | | |
| **Children** | | | | |
| No | 147 (74) | 126 (60) | - | - |
| Yes | 52 (26) | 83 (40) | 1.8 (1.2–2.8)[**] | 1.9 (1.2–2.9)[**] |
| *Missing* | 44 | 3 | | |
| Cat | | | | |
| No | 194 (80) | 146 (69) | - | |
| Yes | 49 (20) | 66 (31) | 1.8 (1.2–2.7)[**] | |
| Missing | 0 | 0 | | |
| Other pet | | | | |
| No | 226 (93) | 193 (91) | - | |

*(Continued)*

**Table 4.** (Continued)

| | Dog owners | | | |
|---|---|---|---|---|
| | No (N = 243)(%) | Yes (N = 212)(%) | Univariable analysis OR (95% CI) | Multivariable model OR (95% CI) |
| Yes | 17 (7) | 19 (9) | 1.3 (0.7–2.6) | |
| *Missing* | 0 | 0 | | |

*p<0.05; *p<0.01;

***p<0.001;

LR = -267.32 (p<0.001)

## Identification & registration rate: Dog Registry

Italy was probably the first country in the world to make dog I&R compulsory in 1991 [7]. Public and private veterinarians are obliged to register new born puppies, owner and address changes and to remove dogs in case of death. Recently, other European countries are implementing DRs, while no DRs are required in the rest of the world [7]. The dog I&R framework in European countries is lagging behind from both a legislative and an operative point of view [7]. In this process, the assessment of the completeness of the I&R systems is a critical requirement.

Researchers exploring the completeness of the DR in Italy found that it varied from 51% in 2004, 55% in 2005 to 71% and 75% in 2011 [9, 17, 28, 32]. As expected, the awareness in dog I&R increased in the first decade after its establishment, but no improvement has been made since as demonstrated in the latest studies. The Bayesian approach used in this study estimated that almost one fourth of living dogs were undetectable. In the DR, the substantial underestimation of the real population was 33%. Interestingly, almost all the non I&R dogs visited a veterinary clinic at least once a year (Table 2), indicating that private veterinarians do not play their role in the I&R control and should be included in the process to enhance the completeness of the DR. Furthermore, it is worth noting that the proportion of dogs with no I&R was homogenously distributed among the age classes (Fig 1), suggesting that the completeness of the DR is not likely to automatically improve in the future, unless further actions are taken.

In the present study, the majority of I&R dogs were purebred dogs, likely because of the owners' awareness of tracing a dog in case an animal become lost. From the analysis of sex, neutering and habitat, it emerged that a dog was less likely identified and registered if male, not neutered and lived in a rural environment. These findings may be explained by owners' wanting to avoid the responsibility of an unwanted litter, or the consequence of potential car accidents and dog biting that could be caused by owned traceable dogs.

Several strategies can be suggested to encourage I&R. Firstly, promoting the responsible dog ownership among the general population [3, 18]. Secondly, enforcing private veterinarians and dog breeders to be an active and co-operative part of the system. Thirdly, establishing an effective monitoring system by competent authorities. Finally, introducing incentives to enhance DR and fines towards owners who do not identify or register dogs.

## Canine population demography

In the present study, the sex ratio favoured male dogs. This diversity is a constant in all similar studies worldwide [17, 27, 29, 31, 32]. Dog owners may believe that male dogs are preferable because they do not produce unwanted puppies, do not attract free roaming dogs during oestrus and can serve as guard dogs, especially those living in rural areas [30].

The overall proportion of neutered dogs in the present study was much greater than the proportion reported in another Italian province [17]. The proportion of neutered female dogs (55%) was significantly higher than males (8%) and it is worth noting that this was the highest rate reported in all the studies considered, except Ireland [14, 17, 29, 30, 33]. In contrast, only 8% of male dogs were neutered, a percentage identical to that observed from Slater *et al* [17], but far lower than the trend recorded in other European countries, which is close to 30% [14, 33]. It is possible that there was a cultural resistance to the dog male castration in the owners of the present study. Another hypothesis was the intention of preserving the protective behaviour in guard dogs in rural areas, but more causes should be investigated. However, educational efforts and economic incentives could be implemented to increase the percentage of neutering in order to reduce the stray dog population, to decrease the risk of sex-related diseases (tumours and pyometra) and the impact of sex-related undesirable behaviour (bites, noises, roaming, aggressiveness), also considering that more than two third of the animals lived in rural areas and outdoors.

The proportion of purebred dogs in the present study was 47%, a proportion comparable to what reported by other studies in different years in Italy [17, 28] suggesting that this ratio is rather constant over time and geographical areas.

From the analysis of the dog source, it emerged that almost half the dogs were acquired as a gift (i.e. without being payed), followed by one fifth of dogs found (a stray animal being adopted) and purchased. A percentage less than 10% of dogs were adopted and born in house. A previous Italian study showed similar results, while an Irish study found a higher proportion of dogs purchased and similar proportion of dogs adopted [14, 17]. The findings of the present study indicated the lack of a relevant trade of dogs, also if the possibility of puppies imported illegally that are likely not going to be I&R should not be overlooked. Moreover, the presence of a significant stray dog population was confirmed (21% of dogs were found) and this finding revealed the urgency to take measures to enhance the low percentage of dogs' adoption from shelters.

The findings of the present study showed almost one dog out of 10 did not attend regularly an annual veterinary visit. It is possible to assume that this population is not subjected to those veterinary cares commonly performed on an annual basis such as vaccination, deworming and health check-up. Data from literature indicates that the proportion of dogs not visited by a veterinarian yearly vary remarkably among the different world areas. Investigations conducted in a central Italian region and in the USA reported a percentage comparable to our findings (11% and 9%, respectively) [17, 34]. Prata found that only 2.7% of pets were not taken to a veterinarian in Portugal during the past year [35]. These values are in contrast with those from studies from Brazil and New Zealand which reported that almost one third of pets did not receive veterinary cares during a one-year period [13, 36]. Pet owners can abstain from regularly taking the pet to a veterinarian for many reasons such as costs, inadequate understanding of the need, negative feelings about subjecting the animal to stress during transportation and examination [17, 35].

In the multivariable analysis, we found that two variables, namely source and kind of feeding, were statistically associated with the outcome. Owners who purchased dogs definitely tend to take more care for the health of their dog. Since the cost of veterinary care seems to be one of the most important concerns for dog owners, it is plausible that people who chose to purchase a dog might have a favourable economic condition, being more prone to spend for veterinary cares. Our findings also suggest that dogs totally or partially fed with commercial feed were more likely to be visited by a veterinarian than those adopting a homemade diet. There are possible motivations to explain this result. Commercial food is more expensive than homemade, owners who spent more money for the food had also more economic means for

veterinary care. Moreover, dogs with health problems are usually on a diet with commercial products for medical reasons (overweight, allergic or kidney problem, etc. . .) and they need frequent veterinary visits to check the health status.

## Ownership profiles

Several studies investigated dog ownership profiles in different countries [13–17, 35, 37]. Differently from these studies, the present study reported the poor presence of reliable predictors for the dog ownership. Only living in rural areas and the presence of children in the HH were associated with the dog ownership. Living in rural areas seems an important predictor since it was identified as a significant factor by several authors [14, 16, 37]. It is possible that people consider important for the dog's well-being having enough space at their disposal and, consequently, they are more prone to own a pet when it can access outdoors. As other authors, we found that the presence of children was associated with the dog ownership. People can consider owning a pet beneficial for children, so it is reasonable that family decision of owning a dog is positively influenced by their presence in the HH [14, 15]. However, this speculation seems to be in contrast with the results from other investigations that showed a negative correlation between dog ownership and presence of children aged less than 10 years or pre-schoolers [15, 16]. It is possible that the age of the children in the HH can influence this choice. Young children need more care than school-aged children and consequently their parents could prefer not to own a dog because they do not have enough time for its needs. Moreover, older children can likely convince their parents regarding acquiring a dog influencing the family decision of owning a dog [14, 15]. Interestingly, dog owners were also more likely to own a cat than people not owning a dog. Cats were popular in the study area [21], showing that the cohabitation of dogs and cats is not considered as a concern, as already demonstrated by Downes *et al*. but different to the results of Murray *et al* [14, 16]. The findings of the present study showed few risk factors identified among the studied variables. Therefore, the determinants of the dog ownership were almost unpredictable and not associated to social, economic or cultural factors.

In conclusion, this is the first study to provide an estimate of numbers and characteristics of a canine population and its ownership in a large metropolitan area of Europe. The results of the present study may be of use to plan monitoring of zoonosis, canine diseases and welfare, to manage the stray dog phenomenon, and to help investigate human-dog interactions in a public health perspective.

## Supporting information

**S1 Table.**
(DOCX)

**S1 Database.**
(XLSX)

**S2 Database.**
(XLSX)

**S1 Questionnaire.**
(PDF)

**S2 Questionnaire.**
(PDF)

## Author Contributions

**Conceptualization:** Andrea Carvelli.

**Data curation:** Andrea Carvelli, Francesca Iacoponi, Roberto Condoleo.

**Formal analysis:** Andrea Carvelli, Francesca Iacoponi.

**Investigation:** Andrea Carvelli.

**Methodology:** Andrea Carvelli, Paola Scaramozzino, Francesca Iacoponi, Roberto Condoleo.

**Project administration:** Andrea Carvelli, Ugo Della Marta.

**Resources:** Andrea Carvelli, Paola Scaramozzino, Ugo Della Marta.

**Supervision:** Andrea Carvelli, Paola Scaramozzino, Ugo Della Marta.

**Validation:** Andrea Carvelli.

**Writing – original draft:** Andrea Carvelli.

**Writing – review & editing:** Andrea Carvelli, Ugo Della Marta.

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
