## [Decision Letter · Decision Letter 0]

9 Jun 2020

PONE-D-20-13213

Size, demography, ownership profile and identification rate of the owned dog population in central Italy

PLOS ONE

Dear Dr. Carvelli,

Thank you for submitting your manuscript to PLOS ONE. After careful consideration, we feel that it has merit but does not fully meet PLOS ONE’s publication criteria as it currently stands. Therefore, we invite you to submit a revised version of the manuscript that addresses the points raised during the review process.

Your manuscript was reviewed by two experts in the field, who have both made some suggestions for modifications prior to acceptance

If you could write a detailed response to reviewers, this will expedite review when it is resubmitted

We look forward to receiving your revised manuscript.

I wish you all the best with your revisions.

Hope you are keeping safe and well in these difficult times.

Kind regards,

Simon Clegg, PhD

Academic Editor

PLOS ONE

Journal Requirements:

3. In your Methods section, please provide additional information about the participant recruitment method and the demographic details of your participants. Please ensure you have provided sufficient details to replicate the analyses such as: a) the recruitment date range (month and year), b) a description of any inclusion/exclusion criteria that were applied to participant recruitment, c) a table of relevant demographic details, d) a statement as to whether your sample can be considered representative of a larger population, e) a description of how participants were recruited, and f) descriptions of where participants were recruited and where the research took place.

Reviewers' comments:

Reviewer's Responses to Questions

**Comments to the Author**

1. Is the manuscript technically sound, and do the data support the conclusions?

Reviewer #1: Partly

Reviewer #2: Yes

2. Has the statistical analysis been performed appropriately and rigorously? 

Reviewer #1: No

Reviewer #2: Yes

3. Have the authors made all data underlying the findings in their manuscript fully available?

Reviewer #1: No

Reviewer #2: Yes

4. Is the manuscript presented in an intelligible fashion and written in standard English?

Reviewer #1: No

Reviewer #2: Yes

5. Review Comments to the Author

Reviewer #1: Regarding availability of all data. The authors state that all data is available in the manuscript itself. I think that the journal is looking for raw data.

This manuscript attempts to better understand and clarify some important information about the dog population in the region around Rome. Part of the challenge is that there are at least 4 objectives: 1. estimate the owned dog population size and demographics; 2. Determine the proportion of dogs not registered with the national registry and see if there are variable which predict this; 3. Predictors of annual veterinary care; 4. Predictors of dog ownership. And the authors spend a lot more time on #2 than the others in both the intro and discussion. The justification and recommendations are rather superficial and simplistic even though the authors are clearly passionate about this topic.

The manuscript is hampered by the need for English language editing. It is quite wordy and repetitive regardless. I would suggest that the authors organize the manuscript according to the 4 objectives, particularly in the methods, results and discussion.

I have some specific comments below; however, the manuscript requires extensive revisions to integrate the 4 objectives as well as in organization, language and clarity of content before I will dive in row by row. The comments I’ve included here are examples of some of the lack of critical thinking and clarity and I hope will help the authors in their revision.

The introduction is focused on identification and registration, an important issue for the reasons outlined. How then does the paragraph beginning on line 74 fit? The other objectives?

Lines 63 and 65: in the former line the authors state that identification and registration are mandatory and in the latter that registration isn’t required. I think this may be a language issue but I’m not sure.

Line 67: Is there a relevant reference for the 2016 earthquake data?

And if there are already data about registration rates in Rome, what does this study add to that information? That should be in the introduction as well.

If this manuscript is also about dog population demographics, how does that relate to the dog registry work and the 3rd and 4th objectives? That should succinctly be included in the introduction.

Line 108: please include a few sentences briefly describing the study area so the reader doesn’t have to find and read another paper to learn this.

Line 109-14: I don’t understand what was done here. I realize the authors were trying to estimate sample size, but it seems to be for two different pieces of work (yet this is one study) and two different confidence intervals. Were the authors looking for a sample size to provide a specific level of precision on the estimates? And since the two sample sizes are quite different which one was used? Only the larger one is needed if that was used.

Line 119: This is a convenience sample which the authors argue is “representative” in the companion cat paper. A few sentences explaining how the respondents were selected is needed here. And in the discussion some justification of why this would be considered representative.

Line 151: excluded dogs > 16 years old but only present data in figure 1 at 13 years and up. Please be consistent and give some explanation for the choices.

Line 152: so, all pairwise combinations of the dog characteristics were tested against each other? Why? And what if the expected frequencies were < 5? At least state how many dog characteristics there were and therefore how many tests were done. And what about multiple comparisons with a p-value of 0.05 overall?

Line 153: what exposure variables were used? I’d like to see a list but a reference to a table or appendix would be ok. State how many there were. And that the other two dependent variables were used as independent variables in the other models.

Line 156: the likelihood ratio test tells you which model fits better in comparing models so that you can decide which variables to include. It doesn’t test whether the final model meets the assumptions for logistic regression. That still needs to be evaluated and included here and in the results.

Results: were all continuous data normally distributed so that the mean is a valid measure of the central tendency? Please add to the methods and adjust to a median if needed.

Line 172: I understand using a Bayesian approach for the estimate but if only 58% of dogs were registered in the sample, then how can the overall estimate be so much higher? Seems like there is something missing here.

Line 179: were these the only dog characteristics that were significantly associated? Please provide actual p-values.

Table 1: given that the subsequent tables have descriptive data for each logistic model this table isn’t needed.

Tables 2-4: please add the actual univariable p-value to the variable row. Then the superscript * will not be needed. You could additionally bold the variables that stayed in the multivariable analysis. Were any interactions considered? There was an association between sex and neuter status, was that examined or a new variable created and used as was shown in Table 1? Why weren’t the owner demographics included in the registry and veterinary care models?

The first part of the discussion feels very generic and not really related to this manuscript and study. Please focus on the objectives of the study and how they relate to important other studies and specifically how they can be used in the future. Order the discussion based on the objectives and don’t include a lot of repetition of results or every article and comparison. This should be a thoughtful and critical discussion of what the results add to the knowledge base, what the limitations are and what can be done in the future. I have only included a few comments below.

Line 234 and following seems like it is related to the limitations of this study? That should be its own section later in the discussion with all other pertinent issues.

Line 253: the term poorer human development index is used several times in the manuscript without definition. And the references here are for countries that are considered to be developed except for references 23 and 24 which might be considered to be poorer or developing. Please edit.

Line 279: reference 26 is a tumor registry not a dog identification and registration paper.

Line 292-3: the authors have a very traditional view that if only the owners were more “responsible” things would be better. That is a very judgmental and often inaccurate and unhelpful portrayal of pet owners. There are real barriers in some cases for owners to do what they should. In fact, alternative reasons to those stated in this sentence could be that they didn’t know they were supposed to register the dog, didn’t know how or what to do, hadn’t had time yet to do it, etc. Please edit.

Reviewer #2: Page 4, line 69: “A number of papers have been published on owned dog population estimate”. I think this sentence seems lost and it can be deleted.

Page 5, line 79: “electronic transponder”. Is this a microchip?

Page 5, lines 84-85: Please briefly explain the main purpose of the DR.

Page 6, Questionnaire design section: How the individuals were selected? Did the authors interview the head of the household or any available individual? I also missed the ethic consideration regarding the interviews.

Page 8, lines 161-164: These are interesting results regarding the human population. However, a demographic description of your human population is missing, and it would add more value to your manuscript. Also, I think authors should provide an additional table for the human demographics.

Page 16, Discussion: I would not write the discussion in topics. Its preferable to be a flowing text as in the background.

The main conclusion and limitations of the study are missing.

6. PLOS authors have the option to publish the peer review history of their article (what does this mean?). If published, this will include your full peer review and any attached files.

Reviewer #1: No

Reviewer #2: No

---

## [Author Response · Author response to Decision Letter 0]

23 Jul 2020

Dear reviewers,

thank you for your comments, which greatly improve the manuscript.

We have followed your instructions in revising the manuscript and put the comments in tables submitted as separate file to facilitate the revision process. One comment regarding the subsections of a reviewer is in conflict with revisions requested by the other one. We apologize but we can not accept both requests.

We also add in the manuscript and in the revision process in PlosONE website, the data availability and the details on interviews as requested by a reviewer.

We hope that, after the performed changes, the article meets the target and the requirements of PLoS ONE.

Sincerely

Andrea Carvelli

---

## [Decision Letter · Decision Letter 1]

15 Sep 2020

PONE-D-20-13213R1

Size, demography, ownership profiles and identification rate of the owned dog population in central Italy

PLOS ONE

Dear Dr. Carvelli,

Thank you for submitting your manuscript to PLOS ONE. After careful consideration, we feel that it has merit but does not fully meet PLOS ONE’s publication criteria as it currently stands. Therefore, we invite you to submit a revised version of the manuscript that addresses the points raised during the review process.

Many thanks for re-submitting your manuscript to PLOS One

It was reviewed by experts in the field, and they have requested some changes be made prior to acceptance.

If you could make these changes and write a response to reviewers, that will greatly expedite revision upon resubmission

I wish you the best of luck with your changes

Hope you are keeping safe and well in these difficult times

Thanks

Simon

We look forward to receiving your revised manuscript.

Kind regards,

Simon Clegg, PhD

Academic Editor

PLOS ONE

Reviewers' comments:

Reviewer's Responses to Questions

**Comments to the Author**

1. If the authors have adequately addressed your comments raised in a previous round of review and you feel that this manuscript is now acceptable for publication, you may indicate that here to bypass the “Comments to the Author” section, enter your conflict of interest statement in the “Confidential to Editor” section, and submit your "Accept" recommendation.

Reviewer #1: (No Response)

Reviewer #3: (No Response)

2. Is the manuscript technically sound, and do the data support the conclusions?

Reviewer #1: Partly

Reviewer #3: Yes

3. Has the statistical analysis been performed appropriately and rigorously? 

Reviewer #1: No

Reviewer #3: Yes

4. Have the authors made all data underlying the findings in their manuscript fully available?

Reviewer #1: Yes

Reviewer #3: Yes

5. Is the manuscript presented in an intelligible fashion and written in standard English?

Reviewer #1: (No Response)

Reviewer #3: Yes

6. Review Comments to the Author

Reviewer #1: The manuscript is understandable with the revisions including the English language edits, but still in need of some editing for English as well as to reduce repetition and make the manuscript more concise, clear and focused. I have therefore not commented line by line, only where I noted obvious needs for editing. Having someone who has written many scientific articles in English language journals review and edit the manuscript would be very helpful!

The Introduction is still too long and a bit repetitive. It moves from national data to Italian registration to international need for data to ownership data in other countries. Please review some recent Plos One articles to see examples of the construction of the introduction. The previous cat focused paper is also an example where the introduction was more focused but don’t use the same sentences--any exact repetition is plagiarism. See as an example Mustiana 2015 in Plos One. Order the topics from most general (international) to most specific (Italy/Rome).

Line 88: note that any fee even a small one, for someone who is very poor may be cost prohibitive. This is an assumption that is untested.

Line 140-2: to be able to say that selection bias was limited, you need to state what was done if the selected person refused to participate here and, in the results, indicate how many people refused to participate.

Page 8: age and number of pets were collected as continuous variables. How did the authors determine what categories to create? Based on what? Gender is applied only to humans. In animals, sex is the preferred term.

Lines 167-8: these sentences are not clear. I think, based on the companion cat study, that the authors are indicating that the interviewers framed the present research as a study of general pet health to avoid bias. And that whether the pet was allowed to roam was not asked?

Line 196: relative to the two sample size calculations, the authors apparently only used the sample size which they first reached, the dog ownership one. Why not also the number interviewed? That needs to be in the manuscript.

Line 209: were the ages normally distributed? Fig 1 would suggest they are NOT normally distributed and therefore the median would be a more correct option.

Table 1 does not appear to add information to that of Table 2. I would either delete or put into the supplement. Alternatively, one column of totals could be added to table 2.

Table 2 only has 312 dogs. What happened to the other 3?

Authors response to L152: does not address my question about multiple comparisons for the chi-square tests for multicollinearity. And the first sentence added is about the chi-square test between outcome and independent variables, not about multicollinearity. Were any variables excluded from any model due to statistical associations with each other? Appears they were not based on a later comment. That needs to be in the results section.

Authors response to L156: in fact, dogs living in the same household are likely not independent which is one of the assumptions for Logistic regression. Please consult with someone who can assist with the statistical analysis.

Authors response to Tables: I’m asking about statistical interactions in the final multivariable models, not multicollinearity. Please add these to the analysis. And because the dogs are clustered or matched within households, households my need to be added to the analysis. That would also allow owner characteristics to be included. There are statistical methods to address this. And if there is a compelling reason to ignore the clustering within households, the authors must include that.

I could make the argument that since the person who answered the survey wasn’t selected as someone who made the decisions for household about the dogs that their characteristics would be unlikely to be related to I&R. That should be included in the discussion as another limitation and may explain why a few HH level variables were significant, but the individual ones were not.

Table 3: please add something to the title about this including the results of 203 owners who were surveyed in x years. And add the 312 dogs to the existing description. Add a bit more to the title, similarly, for Tables 2 and 4.

Lines 271-7: please do not repeat verbatim the section from the cat paper. And there were a number of people selected who refused to participate, no? Please include that in the results. Then adjust this statement and add to the limitations section. This influences non-response and selection biases. Measurement bias wasn’t addressed by the source of participants, only by 1. The training of the interviewers and data entry staff, 2. The validity and reliability of the survey, 3. The way the survey was presented to the respondents, etc. Please be much more careful in the discussion about this and other types of bias and how they were minimized (and if not, how they were likely to influence the results).

Human development index appears in the abstract without any definition or description. Line 292: And isn’t described here either only referenced. Please briefly indicate what indices are included. The next sentence mentions other indices, but none are discussion, unless the dogs per/km2 is the next index—in which case it is about indices to report dog data not potential predictors? Please clarify.

What is the major point of the paragraph starting on line 289? The paragraph ranges through human development index, density in Rome and other countries, and dog and cat ownership.

Line 324: I’m not clear how the homogeneity of the I&R in age (which is quite interesting!) is related to the need for action?

Line 331: Isn’t responsible pet ownership promoted everywhere already? Has it been shown to have an impact on I&R? If there aren’t data to support these strategies, then the authors should at least say how they think these will promote I&R and who and how they should be performed.

Line 343-4: these references are only in a handful of countries and don’t include the US or Canada. Please rephrase.

Line 341: is this about males or neutering in general? The paragraph includes both sexes, but the previous sentences focus on males (I believe).

Line 362: I believe that the authors are saying that if puppies are being imported illegally then they are likely not going to be I&R?

Paragraph starting on line 366 tile 389: There is good information here, but it is long and repetitive. Please edit to be shorter.

Line 406-407: “also more likely to own a cat” than what? This sentence is incomplete.

Lines 409-11: these sentences seem to be saying that: this study was unable to find predictors to confidently predict dog ownership in this or other parts of Italy.

Reviewer #3: This is a nice paper, which has some interesting results from Italy. The science itself seems generally sound, and will be of interest to readers. However, it needs a bit of grammatical reworking, and I have tried to do this below as I am guessing that their first language isn’t English. The changes below don’t need a response writing to them, but hope they help.

Line 19- predictors in the animal population is essential (add in word)

Line 38- The percentage of neutered dogs were higher in females (55%) than in males (8%). (reword)

Line 43/44- including promoting responsible dog ownership, (reword)

Line 57- change human to humans

Line 58- Therefore, the knowledge of the animal population in terms of size (add in word)

Line 62- the lack of information in the pet population is a constant (add in word)

Line 66- owned dog population estimates (make plural, and add in references)

Line 73- please define DRs

Line 75- Nevertheless, while identification is performed on an individual basis, (add in word)

Line 76- change country to countries

Line 78- To date, reliable data on the dog population (add in word)

Line 84- or manage data of both the (add in word)

Line 86- organised toward the general population (add in word)

Line 88- can you please put in the cost of this registration? It may be prohibitive to poor people not to register

Line 90- stakeholders think that the only purpose of the DR is the (add in word)

Line 94- remove high

Line 95- remove the

Line 105- remove an

Line 106- its consequences, such as human and animal injuries (make plural and add in comma)

Line 109- useful for the private sector (add in word)

Line 112- add in comma after evaluated

Line 125- owned dog population, and to identify the human factors…. (change final comma to and)

Line 140- what happened if a person refused to participate? Also, if this was done in 2013- how it is linked to what is relevant now?

Line 141- selecting one in every four people amongst (reword)

Line 147-149- I struggled to follow this, please reword it

Line 155- was asked whether they resided (add in word)

Line 172- estimate the size of the canine population (add in word)

Line 173- comma after ‘investigation (9), ‘

Line 174- remove as prior

Line 74- comma after registered

Line 204- 362,277 dogs were registered in the DR (reword)

Line 210- The age distribution of the dog population showed a moderately different shape for gender (reword)

Line 212- fed mixed food- delete by

Line 216- dogs are usually referred to as gender rather than sex- please change throughout

Table 1 and 2 seem very similar- maybe consider deleting one

Line 234- commonly fed commercial or mixed food (reword)

Line 236- univariable analysis, while multivariable model (add in comma)

Line 258- leishmaniosis- typo- leishmaniasis

Line 260- human bite) and the impact (add in word)

Line 262- demography of the canine (add in word)

Line 290- The percentage of respondents having (make plural)

Line 292- what is the human development index? It would be nice to know what this is and what it means

Line 302- The wide base of the pyramid (add in word)

Line 304- almost identical to that reported worldwide (reword)

Line 305- delete hold

Line 306- replace ‘resulted greater’ with ‘were more common’

Line 310- reword to ‘register new born puppies, owner and address changes’

Line 311- implementing DRs, while no DRs is (add in comma)

Line 315- should be researchers

Line 326- in case an animal becomes lost (reword)

Line 328/9- responsibility of an unwanted litter (add in word)

Line 331- promoting responsible dog ownership (reword)

Line 341- greater than the proportion reported (add in word)

Line 342- reword to – ‘and it is worth noting…’

Line 350- order to reduce the stray dog population (add in word)

Line 360- similar results, while an Irish National (add in comma)

Line 369- commonly performed on an annual basis such as vaccination(add in word)

Line 376- abstain from regularly taking the pet- reword

Line 380- statistically associated with an annual veterinary visit (add in word)

Line 381- Since the cost of veterinary care (add in word)

Line 388- usually on a diet with commercial (add in word)

Line 394/5- identified as a significant factor by (add in word)

Line 396- enough space at their disposal (Add in word)

Line 397- we found that the presence of children (add in word)

Line 407- Cats were popular in the study area… (reword)

Line 408/9 - but different to the results of Murray et al (reword)

Line 410- replace resulted with were

7. PLOS authors have the option to publish the peer review history of their article (what does this mean?). If published, this will include your full peer review and any attached files.

Reviewer #3: No

---

## [Author Response · Author response to Decision Letter 1]

24 Sep 2020

We thank the Editor and the reviewers for their comments, which greatly improve the manuscript.

We have followed your instructions in revising the manuscript and put comments in tables to facilitate the revision process. The file has been uploaded.

---

## [Editor Report · Decision Letter 2]

29 Sep 2020

Size, demography, ownership profiles, and identification rate of the owned dog population in central Italy

PONE-D-20-13213R2

Dear Dr. Carvelli,

We’re pleased to inform you that your manuscript has been judged scientifically suitable for publication and will be formally accepted for publication once it meets all outstanding technical requirements.

Kind regards,

Simon Clegg, PhD

Academic Editor

PLOS ONE

Additional Editor Comments):

Many thanks for resubmitting your manuscript to PLOS One

I have reviewed the manuscript, and as you have addressed all the comments, I have recommended your manuscript for publication.

You should hear from the Editorial Office soon

It was a pleasure working with you, and I wish you all the best for your future research

Hope you are keeping safe and well in these difficult times

Thanks

Simon
---

## [Editor Report · Acceptance letter]

6 Oct 2020

PONE-D-20-13213R2 

Size, demography, ownership profiles, and identification rate of the owned dog population in central Italy 

Dear Dr. Carvelli:

I'm pleased to inform you that your manuscript has been deemed suitable for publication in PLOS ONE. Congratulations! Your manuscript is now with our production department. 

Kind regards, 

on behalf of

Dr. Simon Clegg 

Academic Editor

PLOS ONE